# Paper-Based Microfluidics Perform Mixing Effects by Utilizing Planar Constricted–Expanded Structures to Enhance Chaotic Advection

**DOI:** 10.3390/s22031028

**Published:** 2022-01-28

**Authors:** Chen-Hsun Weng, Pei-Pei Hsu, An-Yu Huang, Jr-Lung Lin

**Affiliations:** 1Medical Device Innovation Center, National Cheng Kung University, Tainan 70403, Taiwan; b88501113@gmail.com; 2Department of Mechanical and Automation Engineering, I-Shou University, Kaohsiung 84001, Taiwan; pphsu@isu.edu.tw (P.-P.H.); kigh666@gmail.com (A.-Y.H.)

**Keywords:** paper-based, mixing, constricted–expanded, chaotic advection, microfluidics

## Abstract

This study aimed to design and fabricate planar constricted–expanded structures that are integrated into paper-based channels in order to enhance their chaotic advection and improve their mixing performance. Chromatography papers were used to print paper-based microfluidics using a solid-wax printer. Three different constricted–expanded structures—i.e., zigzag, crossed, and curved channels—were designed in order to evaluate their mixing performance in comparison with that of straight channels. A numerical simulation was performed in order to investigate the mixing mechanism, and to examine the ways in which the planar constricted–expanded structures affected the flow patterns. The experimental and numerical results indicated that the proposed structures can successfully mix confluents. The experimental results revealed that the mixing indices (*σ*) rose from the initial 20.1% (unmixed) to 34.5%, 84.3%, 87.3%, and 92.4% for the straight, zigzag, curved, and cross-shaped channels, respectively. In addition, the numerical calculations showed a reasonable agreement with the experimental results, with a variation in the range of 1.0–11.0%. In future, we hope that the proposed passive paper-based mixers will be a crucial component in the application of paper-based microfluidic devices.

## 1. Introduction

In recent decades, micro-electromechanical systems (MEMS) technology has been successfully utilized to develop a variety of micromixers, which can be incorporated into microfluidic systems or manipulated as standalone devices. Different micromixers have been widely used in microfluidic applications, such as sample concentration, chemical synthesis, chemical reactors, the polymerization process, biological analysis processes, droplet/emulsion and other processes [1]. However, an effective mixing within microscale devices could be challenging, because laminar flow and molecular diffusion are dominant, given a low Reynolds number. At a microscale, mixing mechanisms are basically cataloged into two approaches for the promotion of mixing effects, namely, streaming advection, and chaotic advection [2]. Streaming (or lamination) advection is produced by setting up multiple channels. On the contrary, chaotic advection is generated by changing the channel geometries or importing external forces. In principle, streaming advection increases the interfacial contact surface area of the confluents. Meanwhile, chaotic advection generates stretching and folding effects to enhance the mixing performance. Hence, the development of effective, reliable, and rapid mixers has become crucial in microfluidics.

Generally, micromixers are classified into two categories—namely, active mixers and passive mixers—depending on the application of external forces. Typically, active micromixers need to import external forces to generate transversal disturbance or chaotic advection. The common external forces include the use of piezoelectrics, pneumatics, acoustics, electroosmosis, dielectrophoresis, magnetohydrodynamics, and electrowetting approaches, which are detailed in the review references [1,3,4]. Such forces facilitate effective mixing and are easy to manipulate; nevertheless, active micromixers suffer from some drawbacks, such as the required additional components that raise fabrication costs, the need for large spaces, and difficult integrability. It is worth mentioning that some newly active micromixers, which had low costs and were simple to fabricate, have been successfully developed to perform effective mixing [5,6,7]. On the contrary, passive mixers mainly rely on the changes in channel geometries instead of employing external forces. Special geometries include 3D serpentine geometries [8], obliquely oriented grooves [9], bas relief structures [10], and modified Tesla structures [11]. When confluents pass through these complex microstructures, streaming advection or chaotic advection is induced to promote the mixing effect passively. Numerical simulations and experimental measurements have demonstrated that these layouts successfully achieve effective mixing [12]. Nevertheless, these special geometries are relatively complicated to fabricate. Moreover, biosamples and particles may cause clogging and fouling problems in dead volumes.

The most common materials of micromixers are silicon, glass, and polymer. Given the rising demand for low costs, disposal alternatives, and biocompatibility in point-of-care diagnostics, paper-based materials have been extensively utilized to fabricate microfluidic devices for biomedical applications. A number of paper-based microfluidic fabrication techniques have been successfully developed in previous works [13,14,15,16,17]. Currently, microfluidic paper-based analytic devices (μPADs) have successfully demonstrated some benefits, including being lightweight, having a low cost, flexibility, portability, and effectiveness. Furthermore, they are used widely in health diagnostics, environmental monitoring, food quality control, drug analytics, and cell assays, etc. [17]. Unlike traditional MEMS-based microfluidic devices, μPADs are easy to fabricate. TheμPADs require hydrophobic barriers, which entail the use of SU-8 [18,19], polydimethylsiloxane (PDMS) [18,20], or wax [21,22]. The fabrication of the hydrophobic barriers involves inkjet printing [23,24], wax printing [21,22,25], and screen printing [15,26], which are technologies that provide low-temperature fabrication processes.

Most importantly, μPADs use intrinsic capillary force to transport fluids without integrating additional micropumps. However, one of the many challenges for μPADs is the development and effective design of mixing samples for reactions and/or detection. This issue presents difficulties because μPADs inherently exhibit a low Reynolds number and uniform flow regimes [18]. Therefore, μPADs should be developed such that they are sufficiently long to ensure adequate mixing. However, an increase in length results in an increase in analysis time. Moreover, fabricating 3D or complicated geometries and importing external forces onto paper-based substrates are difficult. Therefore, a number of studies have exerted great effort to develop paper-based mixers. Thus far, few µPADs have been proposed for application as mixers. For example, Green [27] developed a paper-based passive mixer that incorporates interrupting lines, i.e., rib bone structures, into a Y-shaped channel to successfully promote mixing performance. Rezk et al. [28] proposed paper-based active mixers that apply 30 MHz surface acoustic waves to channels in order to achieve homogenous mixing.

In the current work, the novelties of the proposed paper-based microfluidic mixers are their ease of design, rapid fabrication, low cost, and high mixing performance. The liquids can be both transported and mixed automatically without an additional external source. In addition, a numerical model is used to design the constricted–expanded structures. Moreover, we investigate the flow and concentration fields via a numerical simulation in order to understand the mixing mechanism. Finally, a mixing index is introduced in order to evaluate the mixing performance of the proposed paper-based mixer.

## 2. Theory and Numerical Model

Computational fluid dynamics (CFD) simulations are introduced here in order to numerically investigate the flow and concentration fields within the mixing channels. The modified continuity equation incorporated with the effect of the porous medium is given by
(1)$$\frac{\partial }{\partial t}\left(\varepsilon \rho \right)+𝛻\cdot \left(\varepsilon \rho \vec{U}\right)=0$$
where *ρ* is the fluid density, *U* is the velocity vector of the fluid, and *ε* is the porosity of the medium.

In general, porosity (also known as the pore volume fraction) is defined as the ratio of the void volume (*V_P_*) to the total volume (*V_T_*). For paper, porosity can be derived as [29]
(2)$$\varepsilon =\frac{V_{P}}{V_{T}}=1-\frac{w_{p}}{\rho_{cel}h_{p}}$$
where *w_p_* is the basis weight of the paper (i.e., the ratio of the mass of the paper to the top face area), *ρ_cel_* is the density of cellulose, and *h_p_* is the thickness of the paper.

Alternatively, the Navier–Stokes equation modified by the Forchheimer equation [30] within the porous medium can be changed to
(3)$$\frac{\partial }{\partial t}\left(\varepsilon \rho \vec{U}\right)+𝛻\cdot \left(\varepsilon \rho \vec{U}\vec{U}\right)=-\varepsilon 𝛻P+\varepsilon \vec{B}+𝛻\cdot \left(𝛻\left(\eta \varepsilon \vec{U}\right)\right)-\frac{\varepsilon^{2}\eta }{\kappa }\vec{U}-\frac{\varepsilon^{3}C_{F}\rho }{\sqrt{\kappa }}\left|\vec{U}\right|\vec{U}$$
where *P* is the pressure vector, *η* is the fluid viscosity, *C_F_* is a quadratic drag factor, *B* is the body force vector, and *κ* is the permeability of the porous medium. The last two terms in the equation represent an additional drag force imposed by the pore walls on the fluid within the pores that usually results in a significant pressure drop across the porous medium. In a purely fluid region, the standard Navier–Stokes equation is recovered in order to remove the last two terms.

The Darcy law expresses the relationship between velocity vectors and the pressure in the porous medium; it states that
(4)$$U=-\frac{\kappa }{\eta }𝛻P$$

The capillary pressure is expressed by
(5)$$\Delta P_{C}=\frac{2\gamma_{LG}\cos\theta }{W}$$

Here, *γ**_LG_* is the surface tension, *θ* is the contact angle, and *W* is the channel width.

The modified convection–diffusion equation can be written as
(6)$$\frac{\partial }{\partial t}\left(\varepsilon \rho C\right)+\left(\vec{U}\cdot 𝛻\right)\left(\rho \varepsilon C\right)=𝛻\cdot \left(\rho D𝛻C\right)$$
where *C* is the local concentration and *D* is the diffusion coefficient. Notably, the second term in the left-hand side of Equation (6) represents the advection term, and the term in the right-hand side of Equation (6) represents the diffusion term.

In this study, the flow and concentration inside the channel were numerically simulated using commercial software (CFD-ACEU, CFD-RC, USA) [11,31]. In the simulation process, the modules of flow and chemistry/mixing were adopted to obtain the numerical results. Two parameters—namely, porosity and permeability—were introduced to modify the continuity equation, the Navier–Stokes equation, and the concentration equation under steady-state flow. In all of the simulations, the assumption is that the paper-based microfluidic chip is isothermal, and that the body force effects are negligible. The fluids to be mixed are of constant properties, which includes their density, viscosity, and diffusivity. The physical properties of the fluid and material are listed in Table 1. In this work, the capillary pressure of Δ*P_C_* is calculated to be 425 Nm^−2^, assuming the values of *γ_LG_* = 0.072 Nm^−2^, *θ* = 0, and *W* = 4.0 mm. Hence, in order to simplify the droplet wetting process, the inlet pressure—i.e., the capillary pressure—equals 425 Nm^−^^2^, as the outlet pressures yields 0 Nm^−^^2^. The inlet concentrations of the coloring dye are 1.0 or 0 mM.

## 3. Materials and Methods

### 3.1. Design and Fabrication

We designed four types of analytical µPADs for mixing purposes: straight, zigzag, cross-shaped, and curved channels. The two sample inlets and the one outlet measured 5.0 mm in radius. The flow channel measured 4.0 mm in width and 25.0 mm in length. The rim space of the flow channel was set to 3.0 mm in order to form hydrophobic barriers. Four pairs of planar constricted–expanded structures, i.e., zigzag, cross-shaped, and curved channels, were developed to squeeze and expand the fluids in order to increase the interfacial contact area and generate a transverse effect. When the constricted-to-expanded ratio was 1/4, the effective mixing performance could be achieved by the proposed paper-based mixers. In this study, the constricted gap was limited to 1.0 mm; therefore, the channel width was designed to be 4.0 mm. In summary, the geometric dimensions of every pair for three constricted–expanded structures are detailed in Figure 1.

A schematic flowchart of the fabrication process is shown in Figure 2. We used Autodesk 123D to design the geometry of the μPADs (Figure 2a), and then printed them using a solid-wax ink printer (ColorQube 8570, XEROX, Tokyo, Japan) (Figure 2b). The patterns were wax printed on chromatography paper (No. 51A, TOYO ADVANTEC, Tokyo, Japan). Then, the μPADs were heated to 70 °C on a hotplate for 8–10 min until the wax fully penetrated the paper (Figure 2c). Finally, they were cut into individual paper-based chips for the experimental testing (Figure 2d).

### 3.2. Experimental Testing

The experimental setup comprised a manipulation platform, polytetrafluoroethylene (PTFE) pipettes, and a digital camera (E-5P, Olympus, Tokyo, Japan), as is schematically shown in Figure 3. Each μPADs was placed on an upside-down heat sink to suspend the device in the air. This setup allowed the confluents to flow through the proposed channels without coming into contact with the tabletop. A digital camera was placed above the μPADs for monitoring purposes, and to record the mixing images. For the testing of such paper-based mixers, two water-based colored dyes were used to evaluate the mixing efficiency of the four proposed mixers. The acquired images were processed using ImageJ (v1.8.0, National Institutes of Health, Bethesda, MD, USA) to convert the colors red, green, and blue (RGB) into grayscale values. Finally, the mixing index was calculated by Equation (8) in order to evaluate the mixing performance.

## 4. Results and Discussion

### 4.1. Characterization of the Numerical Simulation

According to Equation (2), we calculated the porosity *ε* = 0.686, given the material properties of the chromatography filter: w_p_ = 87 gm^−2^, t_p_ = 0.18 mm, ρ_cel_ = 1540 kgm^−3^. We maintained a constant porosity of 0.686, and numerically calculated the flow fields against permeability in the range of 4.0–16.0 μm^2^ for the straight channel. Then, we kept a constant permeability of 10^−11^m^2^ and numerically calculated the flow fields against porosities in the range of 0.4–1.0. Figure 4 plots the characterization of the porosity and permeability distributed against the mean velocity for µPADs. The increasing porosity basically decreased the flow velocity, thereby resulting in increased flow resistance. An increase in porosity was found to inversely decrease the fluid velocities; conversely, an increase in permeability was demonstrated to linearly increase the mean velocities. For the comparison of the measurements, the values of *ε* = 0.686 and *κ* = 10 μm^2^ were employed in order to calculate the flow and concentration fields in the later simulations.

The velocity profile is important in the identification of the way in which a mixing phenomenon occurs. The velocity profiles against different porosities appear like a plug, which indicates a uniform streaming flow (Figure 5). A uniform streaming flow implies that transverse velocity did not occur, as it did in a previous study [18]. In other words, the mixing effect does not easily occur in the radial direction. Figure 5 also demonstrates that the mean velocity increases with decreasing porosity. In conclusion, porosity contributes to a uniform streaming flow, but it is not conducive to the radial advection effect.

### 4.2. Characterization of the Mixing Performance

To date, effective mixing within microfluidic devices is still a challenging problem because of the low Reynolds number and uniform flow. For example, the diffusion time (τ) (~*L^2^/D*) is estimated to be approximately 4000 s, assuming a diffusion coefficient and mixing characteristic length of *D* = 10^−^^9^ m^2^ s^−^^1^ [3,28] and *L* = 2.0 mm (i.e., half width), respectively. The mixing time is apparently much longer than expected. Therefore, the promotion of mixing approaches is necessary in order to enhance the mixing effect for microfluidic systems. Herein, red and blue coloring dyes were dropped onto two inlets of a T-shaped junction. These two fluids were induced to flow through the channel via the capillary effect. The average flow times of the straight, zigzag, cross-shaped, and curved channels were measured as 144, 300, 410, and 331 s, respectively. The average velocities of the fluid were 173.6, 83.3, 61.0, and 75.5 μm/s with respect to the straight, zigzag, cross-shaped, and curved channels, respectively. The density, dynamic viscosity, and diffusivity of the solution were assumed to be 1000 kg/m^3^, 10^−6^ m^2^s^−1^, and 10^−9^ m^2^s^−1^, respectively. The paper-based microfluidic channel was designed to measure 25.0 mm in length and 4.0 mm in width. The characteristic mixing length (*L*) was defined as *L* = 4 A_W_/P_W_, where A_W_ and P_W_ denote the wetting area and the wetting perimeter of the flow, respectively. The characteristic mixing lengths were then calculated as 4.0, 2.91, 1.82, and 2.55 mm for the straight, zigzag, cross-shaped, and curved channels, respectively. The Reynolds number (*Re = UL/**ν*) and Péclet number (*Pe = UL/D*) were subsequently converted, as plotted in Figure 6. In this work, the values of Re and Pe were apparently distributed in the range of 0.1–1.0 for the four differently shaped channels. The mixing mechanism belonged to the diffusion-dominated regime, as stated in the reference [3]. In other words, chaotic advection could not be externally induced to disturb the main flow, and the flow field was not conducive to a uniform mix.

We verified the mixing performance of two different colored fluids using the method of color visualization. Red and blue dye droplets, each measuring 20.0 μL, were dropped onto the two inlets of the T-shaped channel. The dyes were slowly transported forward by the capillary effect. Figure 7 and Figure 8 show the concentration distributions for the four mixing channels at the dimensionless times of t* = 0.2 and t* = 1.0. A dimensionless time is defined as t* = t/τ, where t and τ represent the local flow time and the total time, respectively. The images show that the flow was initially laminar and stable because the red or blue coloring fluids were definitely separated near the T-shaped junction, i.e., in front of the first-pair constricted–expanded structure (Figure 7). When the fluids flowed through the zigzag, cross-shaped, and curved channels, significant mixing was observed because of the planar constricted–expanded geometries. As can be observed in Figure 8a, a slightly radial mixing effect occurred for the straight channel. On the contrary, this displayed the enhancement of the mixing performance behind the zigzag, cross-shaped, and curved constricted–expanded regions. As we expected, the purple color appeared obviously, as shown in Figure 8b–d. These results indicated that the three planar constricted–expanded structures—namely, the zigzag, cross-shaped, and curved channels—achieved the highest mixing performance.

### 4.3. Numerical Verification of the Mixing Performance

Numerical simulations were conducted in order to investigate the mechanism of the effect of the constricted–expanded structures on the concentration fields. The red and blue dyes were used to represent the initial concentrations of 1.0 and 0 mM, respectively. The green color indicated the completely mixed concentration of 0.5 mM. Figure 9 shows the concentration contours distributed along with the four different layouts of the passive mixers. Apparently, the concentration distribution of the numerical simulations was the same as that in the experimental observations. Initially, the two colored fluids were distinctly separated in front of the first-pair planar constricted–expanded region. For the straight channel, a slight mixing effect was noted in the central line of the channel, as shown in Figure 9a. However, the confluents achieved a reasonable mixing performance after the third-pair constricted–expanded structures for the zigzag and curved channels (Figure 9b,d), as well as after the second-pair planar constricted–expanded structure for the cross-shaped channel (Figure 9c). In sum, the fluids flowed through the three different four-pair planar constricted–expanded structures relative to the straight channel. Thus, enhanced mixing performance was achieved, particularly in the cross-shaped channel.

In principle, if streaming is parallel to the main flow direction, then it is not useful for the transversal mixing process. Thus, chaotic advection needs to be introduced to the main flow in order to significantly improve the mixing performance. The streamlines are parallel to one another, such that the mixing effect is dominated by the effect of molecular diffusivity, as shown in Figure 10a. However, as the fluids flowed through the three planar constricted–expanded structures, significant curved streamlines occurred, as shown in Figure 10b–d.

According to Bernoulli’s equation in the radial direction, this can be expressed as
(7)$$\rho \frac{V^{2}}{R}=\frac{\partial P}{\partial r}$$

For ∂*P*⁄∂*r* = 0, the streamlines are linear. For ∂*P*⁄∂*r* ≠ 0, the streamlines are curved. ∂*P*⁄∂*r* > 0 implies that the outside pressure is higher than inside pressure, i.e., the phenomena occurred in the expanded region. Conversely, ∂*P*⁄∂*r* < 0 indicates that the outside pressure is lower than the inside pressure, i.e., that the phenomena were observed in the constricted region. In this work, through the constricted structure, the fluids were squeezed in order to increase the interfacial surface area by the effect of ∂*P*⁄∂*r* < 0. Through the expanded structure, the fluids were sprouted to outer curved streamlines because of ∂*P*⁄∂*r* > 0. As the fluid passed through the four-pair constricted–expanded structures, significant chaotic advection occurred as a result of the effect of successive squeezing and expanding. Hence, significant mixing was achieved downstream of the mixing region, as shown previously in Figure 9b–d.

As was mentioned previously, the concentrations in the images were acquired by ImageJ at a downstream location of 21.0 mm from the T-shaped junction. Figure 11 displays the experimental data in comparison with the numerical results in terms of the concentration distributions and radial distance for the four mixers. The results clearly show that the numerical results are in reasonable agreement with the experimental data, with a variation of 5–10%. This outcome confirmed that the proposed numerical simulation can be used for the optimum design of such paper-based mixers.

A mixing index (*σ*) was introduced in order to survey the mixing efficiency of the mixers; it is expressed as [31].
(8)$$\sigma (x)=\frac{\int_{0}^{1}\left|C_{}^{+}(y^{+})-C_{\infty }^{+}(y^{+})\right|dy^{+}}{\int_{0}^{1}\left|C_{i}^{+}(y^{+})-C_{\infty }^{+}(y^{+})\right|dy^{+}}\times 100\%$$

Here, the mixing index of *σ* is with respect to the local normalized concentration of $C^{+}$ distributed along with the radial local normalized distance of *y*+. $C_{i}^{+}$ is the initial condition of 0 or 1.0 for a completely unmixed state, and $C_{\infty }^{+}$ represents the completely mixed state of 0.5. The samples were deemed completely mixed when *σ* = 100%. Conversely, the samples were deemed completely unmixed when *σ* = 0%.

Thus, we calculated the mixing index based on Figure 11 and Equation (8) in order to evaluate the mixing efficiency. The experimental results showed that the mixing index increased from 20.1% (unmixed) to 34.5%, 92.4%, 87.3%, and 84.3% with respect to the straight, cross-shaped, curved, and zigzag channels, respectively. Similarly, the numerical calculations revealed that the mixing index increased from 21.1% (unmixed) to 38.3%, 93.0%, 88.2%, and 85.3% with respect to the straight, cross-shaped, curved, and zigzag channels, respectively. These results showed that the constricted–expanded structures—i.e., the zigzag, curved, and cross-shaped channels—can achieve a good mixing performance. The relationships between the mixing index and the four mixing channels were also explored herein, and the results are shown in Figure 12. The numerical data were deemed completely consistent with the experimental data when located on the dashed line. Under the dashed line, the numerical mixing index was deemed to be higher than the experimental one. When above the dash line, the numerical mixing index was deemed to be lower than the experimental one. In these results, the variation of the mixing index in the numerical simulations relative to the experimental measurements was slightly higher in the range of 1.0–11.0%.

## 5. Conclusions

This study successfully proposed new analytical µPADs that are capable of transporting and mixing fluids. With these devices, sample liquids can be transported and mixed automatically without an additional external source. The proposed µPADs were easily fabricated using wax-printing technology. In addition, a numerical model successfully verified the design feasibility of such mixers on the basis of the comparison of the numerical results with the experimental measurements. The experimental and numerical results showed that the mixing index increased from the initial value of 20.1% (the unmixed state) to 34.5%, 84.3%, 87.3%, and 92.4% with respect to the straight, zigzag, curved, and cross-shaped channels, respectively. The variation of the mixing index in the numerical simulations relative to the experimental measurements was in the range of 1.0–11.0%.

## Figures and Tables

**Figure 1 sensors-22-01028-f001:**
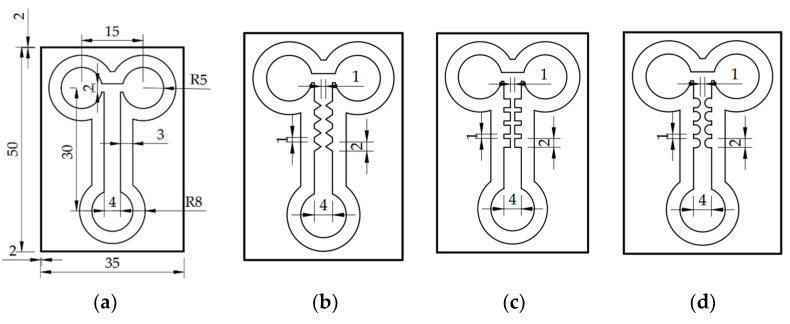
Schematic of the four different geometries: (**a**) straight, (**b**) zigzag, (**c**) cross-shaped, and (**d**) curved channels. Here, the zigzag, cross-shaped, and curved channels were designed with four pairs of constricted–expanded structures to produce the mixing effect. Here, the unit is mm.

**Figure 2 sensors-22-01028-f002:**
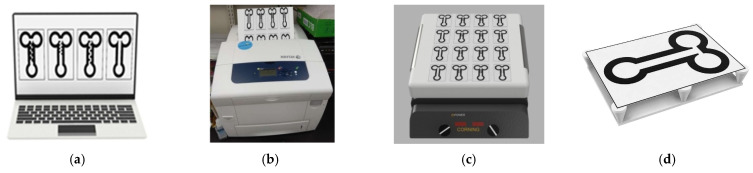
Schematic flowchart of the fabrication process. (**a**) Design, (**b**) wax printing, (**c**) baking, and (**d**) cutting.

**Figure 3 sensors-22-01028-f003:**
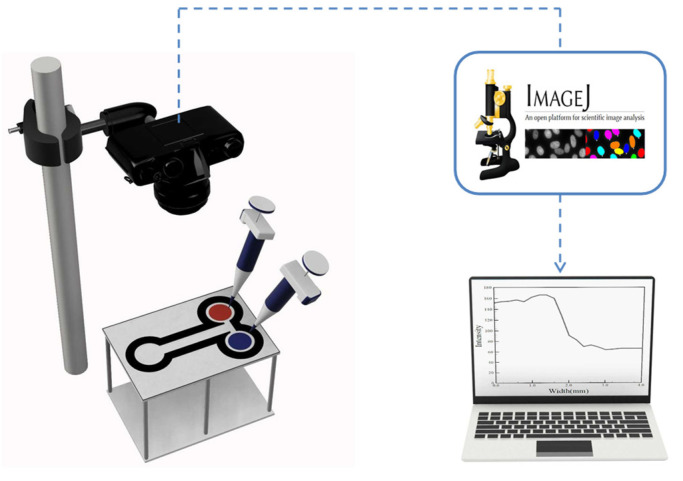
Schematic diagram of the experimental setup.

**Figure 4 sensors-22-01028-f004:**
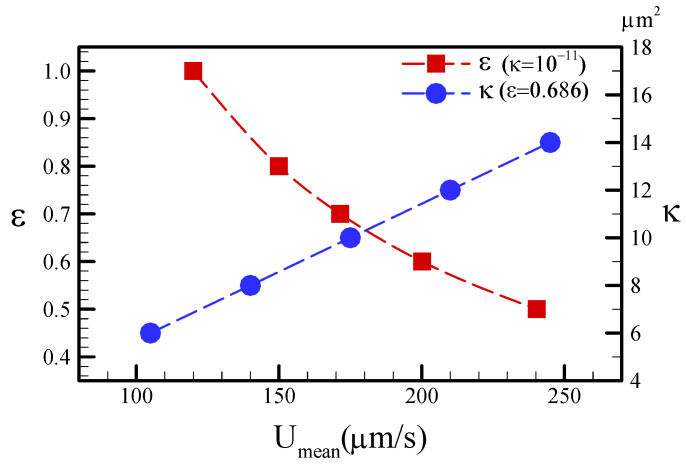
Mean velocity distributed against different porosities (*ε*) and permeabilities (*κ*).

**Figure 5 sensors-22-01028-f005:**
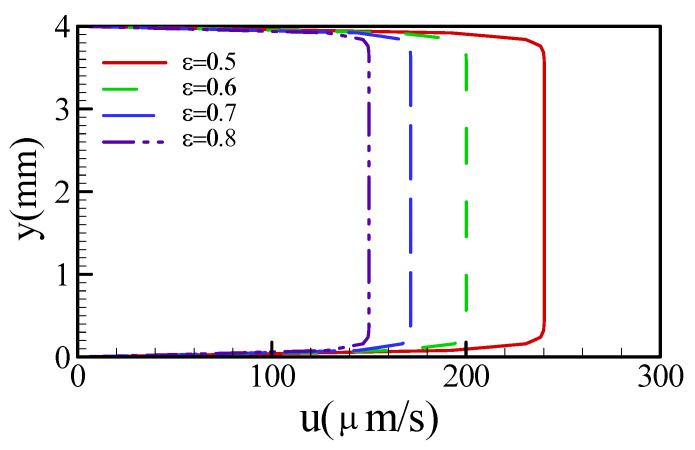
Velocity profiles within the microfluidic channel against different porosities.

**Figure 6 sensors-22-01028-f006:**
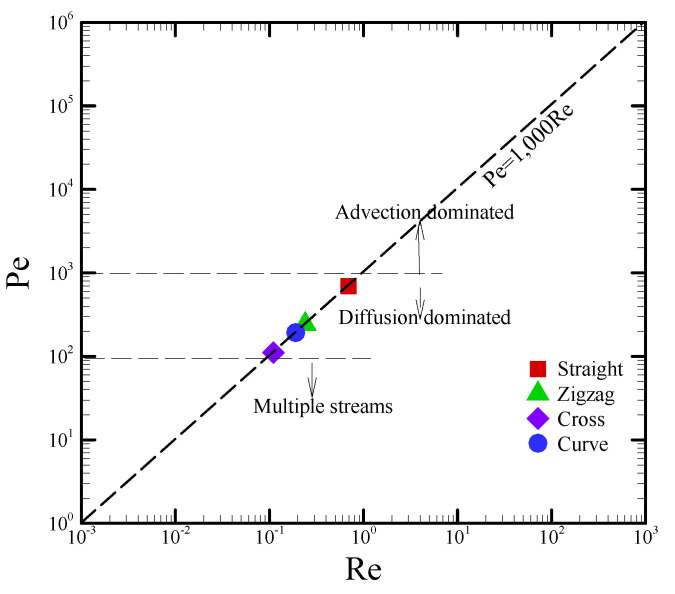
Four differently shaped mixing channels in the Re–Pe diagram.

**Figure 7 sensors-22-01028-f007:**

Images of the coloring dyes within (**a**) the straight, (**b**) zigzag, (**c**) cross-shaped, and (**d**) curved channels for the corresponding time of t* = 0.2.

**Figure 8 sensors-22-01028-f008:**
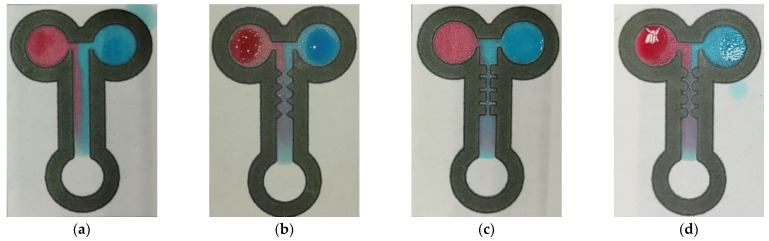
Images of the coloring dyes within (**a**) the straight, (**b**) zigzag, (**c**) cross-shaped, and (**d**) curved channels for the corresponding time of t* = 1.0.

**Figure 9 sensors-22-01028-f009:**
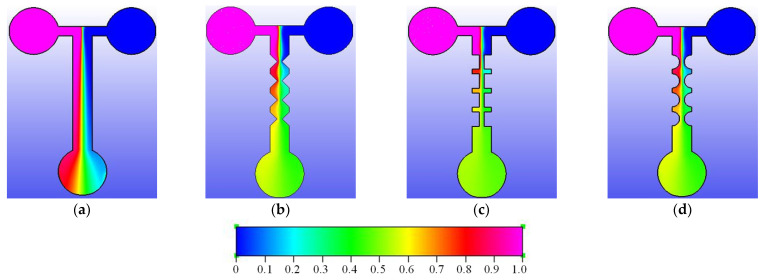
(**a**) Straight, (**b**) zigzag, (**c**) cross-shaped, and (**d**) curved channels, and the corresponding concentration distributions. The legend for the concentrations is displayed below; the unit is mM.

**Figure 10 sensors-22-01028-f010:**
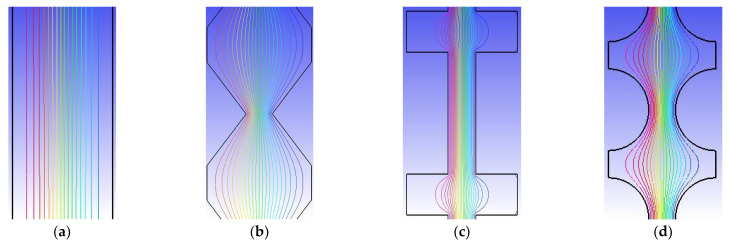
(**a**) Straight, (**b**) zigzag, (**c**) cross-shaped, and (**d**) curved channels, and the corresponding streamline distributions.

**Figure 11 sensors-22-01028-f011:**
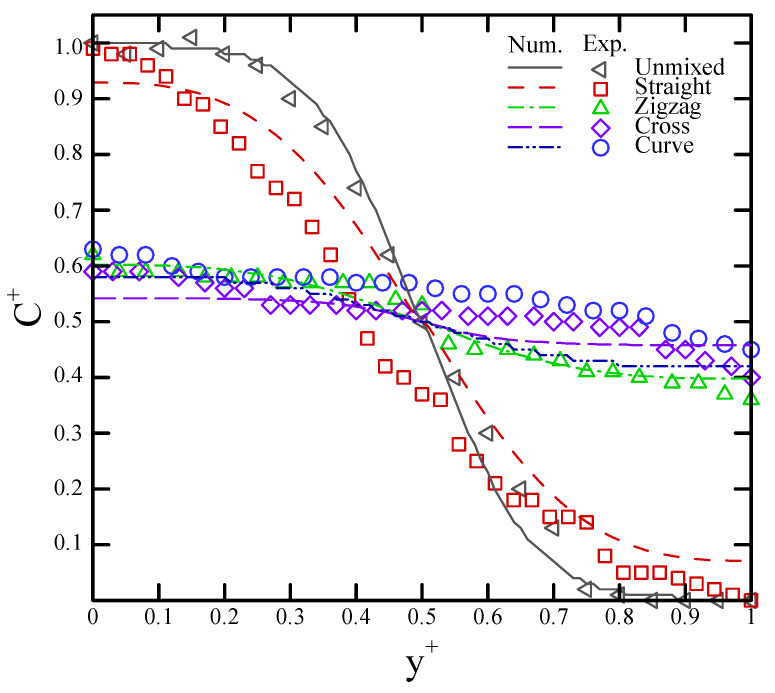
Comparison of the numerical and experimental data in terms of the dimensionless concentration distributions of the four mixers.

**Figure 12 sensors-22-01028-f012:**
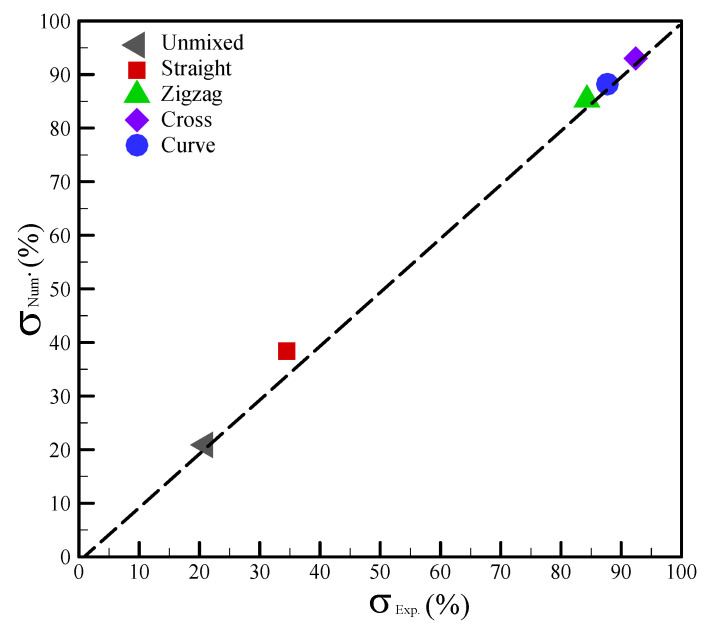
Comparison of the numerical and experimental mixing indices for the differently shaped channels.

**Table 1 sensors-22-01028-t001:** Physical properties of the fluid and material.

Fluid	Property	Symbol	Unit	Value
water	density	ρ	kgm^−3^	1000
	viscosity	μ	Pas^−1^	10^−3^
	diffusivity	D	m^2^s^−1^	10^−9^
	Schmidt number	Sc		10^3^
filter paper	porosity	*ε*		0.686
	permeability	*κ*	m^−2^	10^−11^

## Data Availability

The data presented in this study are available within the article. As for the others that support the findings of this study are available upon request from the corresponding author.

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
