# Peer review of "Paper-Based Microfluidics Perform Mixing Effects by Utilizing Planar Constricted–Expanded Structures to Enhance Chaotic Advection"

_sensors, 2022, doi:10.3390/s22031028_

Round 1
Reviewer 1 Report
The manuscript is about fabricating planar constricted–expanded structures that are integrated into paper-based channels to improve chaotic advection and their mixing performance. The manuscript is well written. I recommend publishing the manuscript. I have a few suggestions to improve the manuscript. 1. The English should be improved.2. More discussion about the novelty of the work.
Reviewer 2 Report
The main voice of this paper is to investigate the performance of four different constricted–expanded structures in paper-based microfluidic mixers. This paper is readable enough and easily we can follow the points of the paper. Numerical results and comparisons are well defined and everything seems working well. Against the above-mentioned positive points, this paper does not experimentally sound, and also there is no remarkable novelty on it. We can find several similar constricted–expanded like structures that they have also a high mixing index. The fabrication method (wax printing) has been already developed and no specific application has been defined as the user of the proposed mixers. I’m not talking about the negative points of lack of novel method, I’m talking about this point that if you have not presented a novel mixing method, at least, you need to focus more on the presented application.
Hence and based on the above considerations, I think this paper needs Major Revisions to be publishable as a journal paper in MDPI sensors. Some other also comments have been listed here:
1- In the introduction part, I think it is better to say something more about active micromixers. I’m on your side and I believe as long as we can use a passive micromixer it is not optimal to use active ones exactly because of the points that you have mentioned in the paper (low cost, complex facilities, point of care applications, etc). But you can find some newly developed active micromixers [1-3] that can be low cost and simple like passive ones (of course not as simple as paper-based). Hence I recommend including these types of active micromixers in the Introduction.
2- Why and how did you propose the straight, zigzag, curve, and cross-shaped structures? Why did not you choose other structures, for example pulsatile, sawtooth, triangular, etc. or a variety of other structures that you can find in the literature? There was any calculation or analysis behind choosing these four structures?
3- The type of your micromixers is too much big. 4 mm length channel is not a proper microfluidic chip.
4- The lack of hardware setup to measure the mixing performance of mixers is totally tangible in the body of the paper.
5- An schematic for the fabrication procedure is recommended.
[1] M. Annabestani, S. Azizmohseni, P. Esmaeili-Dokht, N. Bagheri, A. Aghassizadeh, and M. Fardmanesh, "Multiphysics Analysis and Practical Implementation of a Soft μ-Actuator- Based Microfluidic Micromixer," Journal of Microelectromechanical Systems, vol. 29, no. 2, pp. 268-276, 2020, doi: 10.1109/JMEMS.2020.2975560.
[2] M. Annabestani, H. Mohammadzadeh, A. Aghassizadeh, S. Azizmohseni, and M. Fardmanesh, "Active Microfluidic Micromixer Design using Ionic Polymer-Metal Composites," in 2019 27th Iranian Conference on Electrical Engineering (ICEE), 30 April-2 May 2019 2019, pp. 371-375, doi: 10.1109/IranianCEE.2019.8786743.
[3] C. Meis, R. Montazami, and N. Hashemi, "Ionic electroactive polymer actuators as active microfluidic mixers," Analytical Methods, 10.1039/C5AY01061F vol. 7, no. 24, pp. 10217-10223, 2015, doi: 10.1039/C5AY01061F.
Round 2
Reviewer 2 Report
I have no comments